# Buffer Traps Effect on GaN-on-Si High-Electron-Mobility Transistor at Different Substrate Voltages

**DOI:** 10.3390/mi13122140

**Published:** 2022-12-03

**Authors:** Yuan Lin, Min-Lu Kao, You-Chen Weng, Chang-Fu Dee, Shih-Chen Chen, Hao-Chung Kuo, Chun-Hsiung Lin, Edward-Yi Chang

**Affiliations:** 1Department of Materials Science and Engineering, National Yang Ming Chiao Tung University, Hsinchu 30010, Taiwan; 2Institute of Lighting and Energy Photonics, National Yang Ming Chiao Tung University, Hsinchu 30010, Taiwan; 3Institute of Microengineering and Nanoelectronics, Universiti Kebangsaan Malaysia, Bangi 43600, Malaysia; 4Semiconductor Research Center, Hon Hai Research Institute, Taipei 11492, Taiwan; 5International College of Semiconductor Technology, National Yang Ming Chiao Tung University, Hsinchu 30010, Taiwan; 6Institute of Electronics Engineering, National Yang Ming Chiao Tung University, Hsinchu 30010, Taiwan

**Keywords:** GaN, HEMT, substrate voltage, breakdown voltage, dynamic on-resistance, donor trap, charge redistribution

## Abstract

Substrate voltage (V_SUB_) effects on GaN-on-Si high electron mobility transistors (HEMTs) power application performance with superlattice transition layer structure was investigated. The 2DEG conductivity and buffer stack charge redistribution can be affected by neutral/ionized donor and acceptor traps. As the donor/acceptor traps are excessively ionized or de-ionized by applying V_SUB_, the depletion region between the unintentionally doped (UID)/Carbon-doped (C-doped) GaN layer may exhibit a behavior similar to the p–n junction. An applied negative V_SUB_ increases the concentration of both the ionized donor and acceptor traps, which increases the breakdown voltage (BV) by alleviating the non-uniform distribution of the vertical electric field. On the other hand, an applied positive V_SUB_ causes the energy band bending flattener to refill the ionized traps and slightly improves the dynamic R_on_ degradation. Moreover, the amount of electrons injected into the buffer stack layer from the front side (2DEG channel/Ohmic contact) and the back side (AlN nucleation layer/superlattice transition layer) are asymmetric. Therefore, different V_SUB_ can affect the conductivity of 2DEG through the field effect, buffer trapping effect, and charge redistribution, which can change the electrical performance of the device.

## 1. Introduction

In recent years, GaN-on-Si power HEMTs have gained tremendous interest due to their superior electrical characteristics, such as high breakdown voltage (BV), low on-resistance (R_on_), and fast switching speed [1,2,3]. Compared to GaN-on-SiC and GaN-on-sapphire, GaN-on-Si is most suitable for commercialization owing to the low cost and the scalability to large wafer sizes [3]. The epitaxial structure of a lateral GaN-on-Si buffer stack from bottom to top is usually composed of an AlN nucleation layer, called a superlattice, or graded AlGaN transition layer, a carbon-doped (C-doped) GaN layer, and an unintentionally doped (UID) GaN channel layer [4]. However, GaN buffer stacks usually have plenty of dislocations, lattice defects, and bulk traps due to the lattice and thermal expansion coefficient mismatch between GaN and silicon [5]. The dynamic R_on_ of GaN-on-Si HEMT is related to the surface- and buffer-induced trapping. When the devices are switched from the off- to on-state, the hot electrons generated in the two-dimensional electron gas (2DEG) channel are injected into the surface and buffer layers. When the surface and buffer traps cannot immediately release the trapping electrons, the 2DEG channel is partially depleted to increase the dynamic R_on_ [6].

The surface-induced trapping can be suppressed by the passivation and field plate. In contrast, the buffer-induced trapping caused by the compensated doping and crystalline defects is more challenging to address. During the epitaxial growth, the AlN nucleation layer is usually defective. It contains many donor-type impurities (e.g., Si, O), resulting in a lower effective energy barrier at the AlN/Si junction [7,8] for the enhanced charge injection to occur. Moreover, due to the energy barrier at the AlN–Si substrate interface, the electrons induced from the Si substrate may also be injected into the buffer with positive V_top-to-sub_ (or negative V_SUB_) by trap-assisted tunneling, thermionic emission, and Poole–Frenkel emission [7,8,9].

For the GaN-on-Si HEMT, carbon doped (C-doped) is a common technique to reduce buffer conductivity, reduce current collapse, and increase BV during the epitaxial growth process [10]. The carbon atoms can substitute Ga or N sites, occupy interstitial positions in the nitride, and form chemical complexes with intrinsic defects [11,12,13]. A dominant deep-acceptor trap at E_V_ +0.9 eV (generally considered to correspond to the C_N_ state) and a shallow donor trap at E_C_ −0.11 eV (more likely related to the C_Ga_ state) of the two main energy states are related to the carbon doping in the GaN buffer [10]. An appropriate ratio of acceptor trap and donor trap concentration may exist to achieve the optimum BV and buffer leakage current due to the vertical electric field modulation [14].

C-doped GaN layers could behave as p-type materials due to sufficient compensation doping and the Fermi level (E_F_) in the lower half of the bandgap [15]. UID GaN layers are observed to show light n-type conductivity owing to the existence of nitrogen vacancies [4]. The neutral, ionized donor, and acceptor traps can affect the 2DEG conductivity and buffer-stacks charge redistribution. The applied V_SUB_ or V_D_ at the UID/C-doped GaN interface may result in a behavior similar to the p–n junction [4]. The applied positive/negative V_SUB_ causes different charging/discharging behaviors, which could be related to the non-uniform spatial distribution of dopant traps and the charge redistribution in the buffer stacks. The intrinsic trap time constants and the charge transport in the buffer stacks are both related to the response of donor/acceptor traps in the UID/C-doped GaN layers [15].

Recently, several studies discussed the impacts of V_SUB_ on buffer-related performances, such as dynamic R_on_ [4,16] BV [17,18] vertical leakage current [19,20], gate charge change [21], and output capacitance [22]. However, most works were limited to the individual properties mentioned above. There are few reports [17,18] to investigate the overall device characteristics such as BV, on-state drain current, and dynamic R_on_ respective to V_SUB_ for a GaN HEMT structure, thus, they are lacking unified models and discussions. Compared to these reports [17,18], which applied a V_SUB_ of about ±60 V, we can apply a higher V_SUB_ because the superlattice transition layer with a higher energy barrier can prevent electrons from injecting into the buffer stack layer at a high V_SUB_. In this study, we investigate the effects of a positive/negative V_SUB_ on the above-mentioned device characteristics of the AlGaN/GaN HEMT having a superlattice transition layer. The impact of 2DEG conductivity and the charge trapping/de-trapping procedure are also investigated under the different V_SUB_ and off-state V_D_.

## 2. Materials and Methods

The AlGaN/GaN heterostructures were grown on a 6-inch silicon substrate by metal-organic chemical vapor deposition (MOCVD), as shown schematically in Figure 1. The AlGaN/GaN HEMT structure consists of an AlN nucleation layer, a superlattice transition layer, a C-doped GaN layer, a GaN channel layer, and 22 nm Al_0.25_Ga_0.75_N barrier layer. The mesa isolation was performed by BCl_3_ + Cl_2_ mixed gas plasma etching. To perform the Ohmic contact formation at source and drain, Ti/Al/Ni/Au was deposited by e-beam evaporation and was annealed by rapid temperature annealing (RTA) at 840 °C for 30 s in N_2_ ambient. The Ni/Au (50/300 nm) Schottky gate contact was deposited by the electron beam evaporation. Finally, 20 nm SiN was deposited as a passivation layer. The source-drain length, gate length, and gate width were maintained at 20, 3, and 25 μm, respectively. The buffer stacks can be considered insulated capacitors when the leakage current of each layer is smaller than the “displacement current”; i.e., I_DISS_ = C_TOT_dV_SUB_/dt, C_TOT_ is the series combination of the capacitances of UID GaN, C-doped GaN, and superlattice transition layer, respectively [15].

## 3. Results and Discussion

As shown in Figure 2, the buffer trap charging/discharging mechanism of HEMT devices with a superlattice transition layer is demonstrated schematically. The substrate of GaN-on-Si power HEMT can be used as an independent contact terminal for applying V_SUB_ [23]. The V_SUB_ can modulate the 2DEG concentration through the back-gate effect. Figure 2a shows that the buffer acceptor and donor traps are partly ionized and neutralized at the initial state. The un-ionized neutral donor/acceptor traps may exist in the UID/C-doped GaN layer. The opposite doping polarities may lead to the formation of the p–n junction at the UID/C-doped GaN interface [4,9,15].

Figure 2b shows that the electrons are injected from the Ohmic contact or 2DEG channel into the buffer layer when applying a positive V_SUB_ [24,25]. The Ohmic contact provides the reservoir of free electrons that can be injected into the buffer layer when applying a positive V_SUB_. Without applying V_SUB_, the acceptor trap pulls down the E_F_ in the buffer/transition layer so that the E_F_ is below the acceptor trap level (E_TA_). Many acceptor traps are not ionized (charge-neutral) because the acceptor traps are deep. When the positive V_SUB_ increases, the high vertical electric field induces electron trapping (ionization at the acceptor trap level) in the buffer layer of the acceptor trap [24,25]. Because a positive V_SUB_ of 200 V is smaller than the traps-filled-limit voltage of donor traps (V_TFL2_), both with the positive V_SUB_ and without the V_SUB_ do not affect the concentration of the ionized donor trap.

Figure 2c shows that the concentration of both ionized acceptor and donor traps increases with the negative V_SUB_. In addition to the inherent barrier of the AlN/Si junction, the superlattice transition layer makes the electrons more difficult to be injected into the buffer layer in the off-state under the high V_D_. The top and bottom asymmetric vertical leakage and the fundamental difference between the carrier injection/transport mechanisms induced a different buffer layer trapping. The acceptor traps in the carbon-doped GaN are mostly ionized at a V_SUB_ of −100 V, resulting in a net negative charge [4]. This net negative charge lifts the buffer energy band, increasing the energy barrier of the transition layer to a higher value, and injecting fewer electrons to the buffer layer [16]. The transition layer with a superlattice buffer structure can block the electron injections from the AlN/Si junction to the buffer layer. At off-state and high V_D_, the electrons from the Si substrate begin to overcome the energy barrier of the superlattice layer and inject into the buffer layer. When part of the buffer layer starts to conduct, the negative charge accumulates in the buffer layer, resulting in an increase in the buffer leakage current [15].

Figure 3 shows the energy band diagram when a different V_SUB_ and low/high V_D_ are applied. Figure 3a shows that without V_SUB_, the energy band has more bending as the V_D_ increases. Figure 3b shows that at low V_D_, the applied positive V_SUB_ causes the Si substrate to bend downward. The 2DEG channel tends to inject more electrons into the buffer stack, resulting in an entirely negative charge redistribution in the buffer layer and an increase in the concentration of the ionized acceptor trap. The positive V_SUB_ can alleviate the energy band bending caused by applied high V_D_, and the electrons in the inversion layer formed at the Si substrate/AlN interface are less able to overcome the energy barrier of AlN and superlattice and inject into the buffer layer, avoiding the accumulation of negative charge in the buffer layer. Figure 3c shows that the negative V_SUB_ can raise the energy band of the Si substrate even more and accumulate electrons in the inversion layer at the Si substrate/AlN interface at lower V_D_. After the electrons overcome the AlN energy barrier, the electrons are blocked by the superlattice layer and are not injected into the buffer layer. When the V_D_ becomes larger, the bending of the energy band becomes more significant, which is equal to the applied drain voltage of |V_D_ + V_SUB_| to the device. This causes the electrons accumulated in the Si substrate/AlN inversion layer to have higher energy to overcome the energy barrier of AlN and superlattice to inject into the buffer layer. This results in a significant amount of electrons injected into and accumulated in the C-doped GaN layer.

Table 1 and Figure 4 compare the overview of DC characteristics under the positive/negative V_SUB_. Applying the positive V_SUB_ tends to induce more electrons in the 2DEG channel. However, the maximum drain current (I_D_) is almost constant because the ionized acceptor traps caused by injection electrons from the 2DEG channel or ohmic contact could clamp the increasing 2DEG concentration [4]. Under the positive V_SUB_, the DC characteristics are similar to the device without V_SUB_. A slightly reduced off-state I_D_ is observed with a positive V_SUB_ due to the leakage current flow from the drain to the buffer layer. Part of the electrons are also captured by the acceptor trap of the GaN carbon-doped layer, resulting in a slightly reduced total leakage current. Figure 2b shows that due to the applied positive V_SUB_, the electrons from the Ohmic contact at the interface of source and drain with GaN or 2DEG are injected into the buffer layer, where the acceptor trap of the GaN carbon-doped layer is ionized for the lower V_D_ (12 V) case. An applied negative V_SUB_ ideally reduces the maximum I_D_ due to the simple electric field effect or capacitive coupling [15]. The ionization of acceptor traps in the buffer layers depletes the 2DEG channel; both can degrade the 2DEG conductivity. The V_th_ increases because the required gate voltage (V_G_) for turning on the depleted channel is increased. As Figure 2c shows, off-state applied negative V_SUB_ causes the ionized donor trap concentration in the UID GaN layer to increase. An increase in ionized donor trap concentration indicates that positive charges have accumulated in the buffer layer. Hence, the positive charges neutralize part of the leakage current flowing into the buffer layer. Therefore, the total off-state I_D_ decreases, and the on/off current ratio increases. The subthreshold swing (S.S) of this HEMT device is related to the I_D_ magnitude when V_G_ is smaller than the threshold voltage (V_th_). As the off-state leakage current reduces due to the negative V_SUB_, a reduction in S.S is observed. The transconductance (g_m_) hike of this superlattice HEMT device with a negative V_SUB_ applied is also observed. Firstly, the applied negative V_SUB_ reduced the off current and increased the change in current per unit voltage, making the g_m_ maximum larger. Secondly, when the negative V_SUB_ is between −50 and −200 V, charge redistribution occurs in the GaN carbon doping layer. The higher maximum g_m_ is related to the buffer stack charge redistribution due to the capacitance change of UID GaN, C-doped GaN, and superlattice transition layer [26].

Figure 5 shows the BV characteristics measured using off-state leakage current with the positive/negative V_SUB_. Figure 5a,b show that the negative V_SUB_ can increase the BV due to the increased ionized donor trap concentration. The BV was measured starting from V_D_ = 0 V and gradually increased in voltage by 2 V/s. The electrons forming in the inversion layer between the Si substrate and AlN layer slowly accumulate energy to overcome the energy barriers to inject into the buffer stack. The electrons injected into the buffer layer have a sufficient reaction time to neutralize the ionized donor trap in the UID GaN with a positive charge. The ionized donor trap is more critical to BV than the ionized acceptor trap due to the alleviation of the non-uniform distribution of the high electric field caused by high V_D_ [14]. Applying the negative V_SUB_ (or positive V_top-to-sub_) causes the donor/acceptor traps to ionize at the UID/carbon-doped GaN interface [27]. An increase in ionized donor trap concentration with a moderate acceptor trap can increase the BV and reduces the off-state leakage current [14]. Therefore, the concentration of ionized donor trap and BV increase when a negative V_SUB_ is applied. On the other hand, Figure 2a,b show that the electrons from the 2DEG channel or the Ohmic contact are primarily injected into the buffer layer to increase the ionized acceptor traps without decreasing the concentration of the ionized donor traps [15]. The BV at the applied positive V_SUB_ and V_SUB_ = 0 are very similar due to the same concentration of ionized donor traps.

The dynamic R_on_ characteristics for the HEMT devices with different V_sub_ were also measured. After each dynamic R_on_ measurement, the device’s initial state is fully recovered by shining microscopic light for 30 s, so that all trapped electrons were de-trapped [28]. Figure 6a shows that the device is switched to the on-state with a total switching time of 200 μs after 10 ms of high V_D_ stress at off-state. As shown in Figure 6b, the positive V_SUB_ slightly reduces the dynamic R_on_, while the negative V_SUB_ significantly increases the dynamic R_on_. The dynamic R_on_ ratio for applying the positive V_SUB_ is slightly lower than without V_SUB_ (the off-state V_D_ is from 100 to 300 V). As shown in Figure 3b, the positive V_SUB_ can relax the energy band bending of the Si substrate/AlN junction caused by high V_D_, reducing the electrons that can overcome the energy barrier injected into the buffer layer.

Although an ionized acceptor trap in a C-doped layer may screen the electric field of a positive V_SUB_, applying a positive V_SUB_ slightly increases the 2DEG conductivity by the field effect. Due to the electron injections into the buffer layer from the 2DEG channel and Ohmic contact, the ionization of the acceptor trap concentration in the C-doped GaN layer increases by applying a positive V_SUB_. The donor trap concentration is almost similar when applying a positive V_SUB_ and without V_SUB_. The literature has reported that the ionization donor trap in UID GaN can alleviate the dynamic R_on_ degradation [9,14]. The ionization donor trap leads to a redistribution of the positive charge in the buffer stack. However, as shown in Figure 3c, the negative V_SUB_ aggravates the Si substrate energy band bending, allowing the electrons to have higher energy to overcome the energy barriers between the AlN nucleation layer and superlattice transition layer injection into the buffer layer. The electrons injected into the buffer layer are not neutralized by the ionization donor trap (positive charge) in a short time. The 2DEG channel may be depleted, resulting in dynamic R_on_ degradation. Therefore, under the negative V_SUB_, the dynamic R_on_ degraded more seriously.

## 4. Conclusions

The effect of V_SUB_ on the power application characteristics of GaN-on-Si high electron mobility transistors (HEMTs) with a superlattice transition layer, including breakdown voltage (BV) and dynamic on-resistance (R_on_), was investigated. The negative V_SUB_ increases the BV due to the increasing ionized donor-trap concentration. However, the positive V_SUB_ and without V_SUB_ do not affect the ionization donor-trap concentration, so both BVs are similar. The applied positive V_SUB_ can relax the energy band bending of the Si substrate/AlN junction caused by the high vertical electric field. Therefore, the reducing electrons inject the buffer stack layer. The dynamic R_on_ ratio can be slightly reduced by applying a positive V_SUB_ of 200 V at high off-state V_D_. Although the applied negative V_SUB_ increases the ionized donor trap concentration, it also raises the Si substrate energy band. More electrons overcome the energy barrier and are injected into the buffer layer, causing the buffer trap to capture more electrons, and depleting the two-dimensional electron gas (2DEG) channel, making the dynamic R_on_ degradation more serious. When applying the negative V_SUB,_ S.S decreases, the on/off current ratio increases, and threshold voltage (V_th_) shifts to positive due to the reduction in source-drain leakage current. When applying the positive V_SUB_, the DC performances of the devices, such as on/off current ratio, S.S, and V_th_, remain almost the same as without an applied V_SUB_ because both 2DEG conductivities remain nearly the same.

## Figures and Tables

**Figure 1 micromachines-13-02140-f001:**
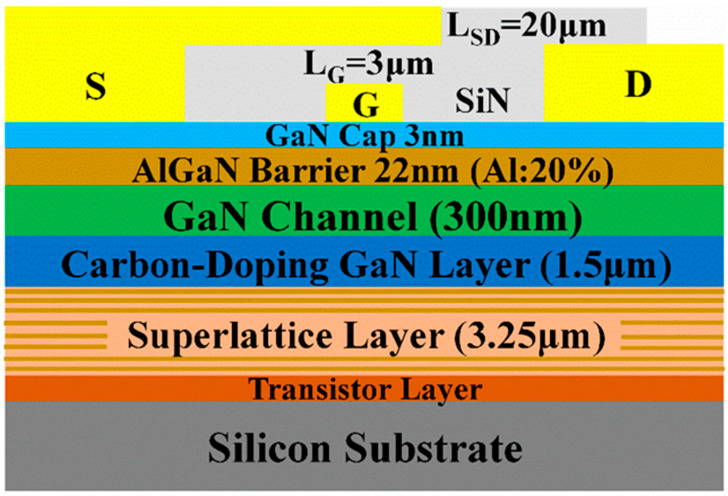
Schematic cross section of the GaN-on-Si device with superlattice buffer.

**Figure 2 micromachines-13-02140-f002:**
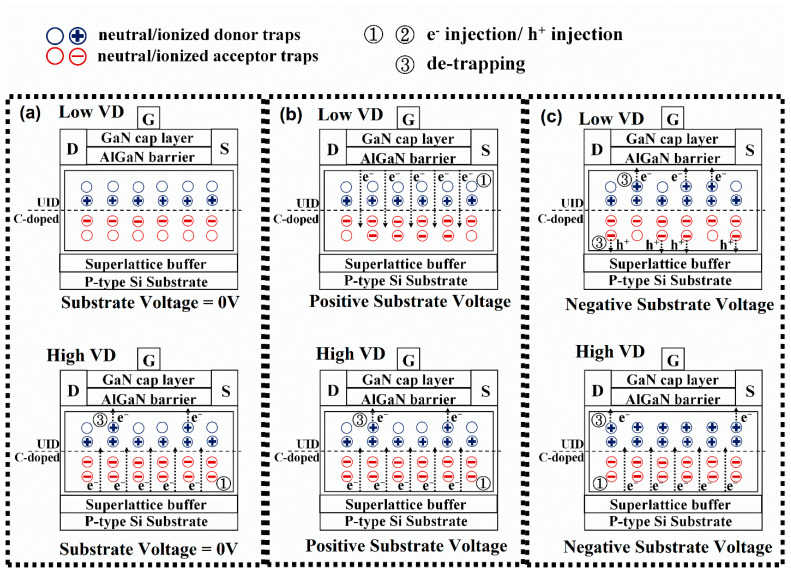
(**a**) Without V_SUB_: the traps charging/discharging in the buffer while applying the low or high drain voltage (V_D_). (**b**) With positive V_SUB_: the traps charging/discharging in the buffer while applying the low or high V_D_. (**c**) With negative V_SUB_: the traps charging/discharging in the buffer while applying the low or high V_D_.

**Figure 3 micromachines-13-02140-f003:**
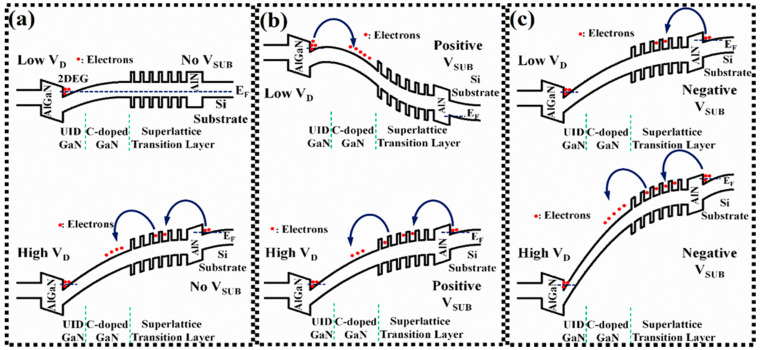
Band diagram with different V_SUB_ and low/high V_D_ condition. (**a**) Without V_SUB_, (**b**) with positive V_SUB_, (**c**) with negative V_SUB_.

**Figure 4 micromachines-13-02140-f004:**
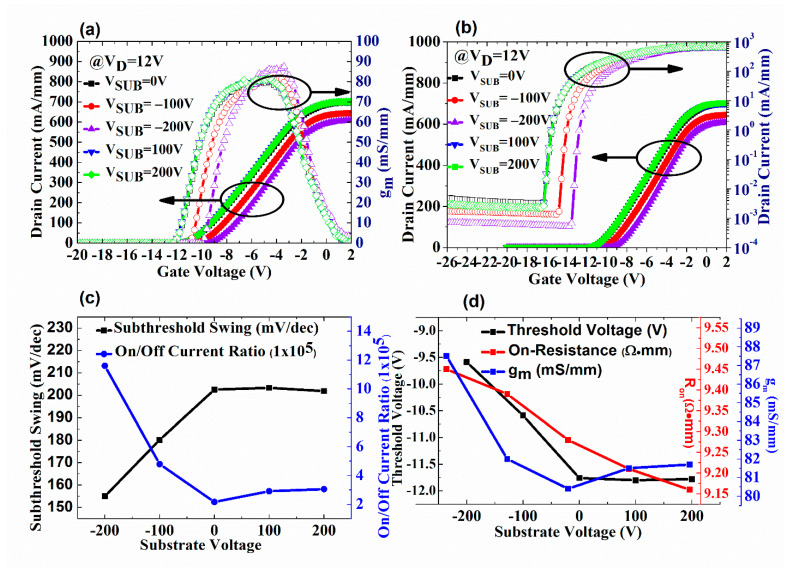
(**a**,**b**) I_D_-V_G_ characteristics of the superlattice-based HEMT for different V_SUB_. (**c**) Comparisons of drain current (I_D_), on/off ratio, and subthreshold swing versus (S.S) different V_SUB_. (**d**) Comparison of V_th_, R_on_, and g_m_ versus different V_SUB_.

**Figure 5 micromachines-13-02140-f005:**
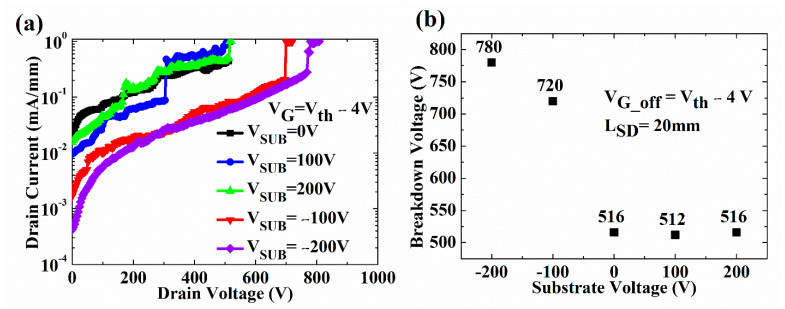
(**a**) Off-state drain leakage current of the GaN-on-Si HEMT as a function of substrate voltage in the log scale. (**b**) BV at different V_D, off_ as a function of various V_SUB_.

**Figure 6 micromachines-13-02140-f006:**
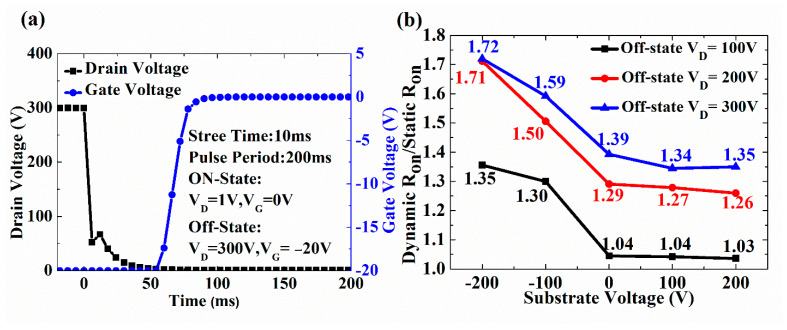
(**a**) Dynamic R_on_ transients of Drain/Gate voltage (V_G_) vs. time (μs) (**b**) Dynamic R_on_/static R_on_ ratio at different off-state V_D_ as a function of various V_SUB_.

**Table 1 micromachines-13-02140-t001:** DC characteristics of the GaN-on-Si HEMT with different substrate voltages.

Substrate Voltage (V)	On/Off I_D_ Ratio	S.S (mV/dec)	V_th_ (V)	R_on_ (Ω•mm)	g_m_ (mS/mm)	I_D,max_ (mA/mm)	Off-State I_D_ (mA/mm)
−200	1.16 × 10^6^	155	−9.59	9.45	87.5	608	5.24 × 10^−4^
−100	4.79 × 10^5^	180	−10.59	9.39	82	644	1.34 × 10^−3^
0	2.16 × 10^5^	202	−11.76	9.28	80.4	696	3.22 × 10^−3^
100	2.91 × 10^5^	203	−11.8	9.21	81.5	702	2.41 × 10^−3^
200	3.05 × 10^5^	201	−11.78	9.16	81.7	701	2.3 × 10^−3^

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
