# Peer review of "Buffer Traps Effect on GaN-on-Si High-Electron-Mobility Transistor at Different Substrate Voltages"

_micromachines, 2022, doi:10.3390/mi13122140_

Round 1

Reviewer 1 Report

The main question addressed by the research is the effect of substrate voltage (Vsub) in a HEMT. And this paper presents what a certain vertical electric field produces in the HEMT in question. It adds material to the subject area compared with other published. I think this article is publishable. The conclusions are consistent with the evidence and arguments presented and they address the main question posed. The references are appropriate. But the text between lines 120 and 121 should be improved. 

Author Response

Dear Review1
Thank you for your comments.

Please see the attachment
Response to Reviewer 1 Comments

Point 1: The main question addressed by the research is the effect of substrate voltage (Vsub) in a HEMT. And this paper presents what a certain vertical electric field produces in the HEMT in question. It adds material to the subject area compared with other published. I think this article is publishable. The conclusions are consistent with the evidence and arguments presented and they address the main question posed. The references are appropriate. But the text between lines 120 and 121 should be improved.

Response 1: Thank you for this comment. We have rectified it. We modify this text between lines 120 and 121 of two sentences. “Because applied a positive VSUB of 200 V is smaller than the traps-filled-limit voltage of donor traps (VTFL2), both with the positive VSUB and without VSUB do not affect the concentration of the ionized donor trap.”

Best wish
Yuan Lin

Reviewer 2 Report

The work of this manuscript is practical and logical. The author has investigated the effects of substrate voltage (VSUB) on AlGaN/GaN HEMTs characteristics, including breakdown voltage (BV) and dynamic on-resistance (Ron), by unified models and discussions. Found that the negative VSUB could increases the BV due to the increasing ionized donor trap concentration, and the dynamic Ron ratio can be slightly reduced by applying a positive VSUB of 200 V at high off-state VD. This manuscript is well arranged with rich content. However, to further improve this manuscript, I have the following suggestions:

1. The motivation for this work should be stated in detail.

2. The significance of this manuscript is not expounded sufficiently. The author needs to highlight this manuscript's innovative contributions.

3.In Fig. 2b, I wonder to know why there is no electrons injected from the ohmic contact or 2DEG channel into the buffer layer when applying High VD, which was different from the marks in Fig. 3(b)-High VD.

4. In Fig. 2c (Low VD), should the source of h+ be modified to ③?

5. The quality of English needs improving.

Overall, it is a meaningful work and is possible to be published after fully consideration of afore mentioned issues.

Author Response

Dear Review2
Thank you for your comments.
Please see the attachment.

Response to Reviewer 2 Comments

The work of this manuscript is practical and logical. The author has investigated the effects of substrate voltage (VSUB) on AlGaN/GaN HEMTs characteristics, including breakdown voltage (BV) and dynamic on-resistance (Ron), by unified models and discussions. Found that the negative VSUB could increases the BV due to the increasing ionized donor trap concentration, and the dynamic Ron ratio can be slightly reduced by applying a positive VSUB of 200 V at high off-state VD. This manuscript is well arranged with rich content. However, to further improve this manuscript, I have the following suggestions:

Point 1: The motivation for this work should be stated in detail.

Response 1: Thanks for your comment. We have added the detail of motivation in the manuscript of the introduction. “And compared to these reports [17, 18], which applied VSUB of about ±40V, we can apply a higher VSUB because the superlattice transition layer with a higher energy barrier can prevent electrons from injecting into the buffer stack layer at high VSUB.”

Point 2: The significance of this manuscript is not expounded sufficiently. The author needs to highlight this manuscript's innovative contributions.

Response 2: Thanks for the comment. We have modified the first sentence of the conclusion of this manuscript to highlight this innovative contribution as  “The effect of substrate voltage (VSUB) on the power application characteristics of GaN-on-Si high electron mobility transistors (HEMTs) with superlattice transition layer, including breakdown voltage (BV) and dynamic on-resistance (Ron), was investigated.”

Point 3: In Fig. 2b, I wonder to know why there is no electrons injected from the ohmic contact or 2DEG channel into the buffer layer when applying High VD, which was different from the marks in Fig. 3(b)-High VD.

Response 3: Thanks for your comment. We have modified both Fig 2(b) and Fig 3(b) that when applied high substrate voltage (VSUB), the ohmic contact or the 2DEG channel would inject the electron into the buffer layer. The energy band diagram would be like a see-saw when we apply fixed positive VSUB and different VD. Three situations could happen:  
  Situation A.: VD is smaller than positive VSUB

    Suppose the positive VSUB minus the VD is bigger than the traps-filled-limit voltage of acceptor traps (VTFL1). The fermi level (EF) would be above the acceptor trap level, and the ohmic contact or 2DEG channel would provide the reservoir of free electrons injected into the buffer layer.

    Situation B.: VD is closed to positive VSUB

    Whether the ohmic contact or 2DEG channel injects electrons into the buffer layer depends on whether the EF is above the acceptor trap level. If the VSUB minus the VD is bigger than VTFL1, the ohmic contact would inject the electron into the buffer layer. At this time, the GaN energy band bending would not be obvious.

   Situation C.: VD is bigger than position VSUB

    The energy band at the ohmic contact side bends downward due to high VD. The electrons would inject from the inversion layer, which formed at the Si substrate/AlN later interface.

  According to the VD measurement range, we indicated that both Fig. 2(b) and Fig. 3(b) are situation A and situation C. So we change Fig. 3(b) at high VD as an ohmic contact side band bending downward, and the electrons are injected from the junction between the Si substrate and AlN layer. And ohmic contact or 2DEG would not inject electrons into the buffer layer.

Point 4: In Fig. 2c (Low VD), should the source of h+ be modified to ③?

  Response 4: Thanks for your comment. We have modified the source of h+ as “â‘¢ de-trapping” in Fig 2(c) Low VD.

Point 5:  The quality of English needs improving.

Response 5: Thanks for your comment. We have modified the quality of the English in this manuscript. Please see the revised manuscript.

Best wish
Yuan Lin

Round 2

Reviewer 2 Report

Current version is almost ready for publication.

An important but minor revision is suggested that all the figures should be shown with high resolution, uniform format, and aligned insets.

Author Response

Response to Reviewer 2 Comments

Current version is almost ready for publication.

An important but minor revision is suggested that all the figures should be shown with high resolution, uniform format, and aligned insets.

Response 2: Thank you for your comment.

A. We have enlarged font size of Fig 3, removed left side "EF", and increased width of Fig 3 and Fig 2 from 16.5 to 17.5 cm.

B. We change Fig 4 (d) up border line to thin.